# Obesity-Related Neuroinflammation: Magnetic Resonance and Microscopy Imaging of the Brain

**DOI:** 10.3390/ijms23158790

**Published:** 2022-08-08

**Authors:** Anita Woo, Amy Botta, Sammy S. W. Shi, Tomas Paus, Zdenka Pausova

**Affiliations:** 1The Hospital for Sick Children, University of Toronto, Toronto, ON M5G 1X8, Canada; 2Departments of Physiology and Nutritional Sciences, University of Toronto, Toronto, ON M5S 1A1, Canada; 3Centre Hospitalier Universitaire Sainte-Justine, University of Montreal, Montreal, QC H3T 1C5, Canada; 4Departments of Psychiatry of Neuroscience, Faculty of Medicine, University of Montreal, Montreal, QC H3T 1J4, Canada; 5Departments of Psychology and Psychiatry, University of Toronto, Toronto, ON M5S 1A1, Canada; 6ECOGENE-21, Chicoutimi, QC G7H 7K9, Canada

**Keywords:** obesity, neuroinflammation, Alzheimer’s disease, magnetic resonance imaging, microscopy

## Abstract

Obesity is a major risk factor of Alzheimer’s disease and related dementias. The principal feature of dementia is a loss of neurons and brain atrophy. The mechanistic links between obesity and the neurodegenerative processes of dementias are not fully understood, but recent research suggests that obesity-related systemic inflammation and subsequent neuroinflammation may be involved. Adipose tissues release multiple proinflammatory molecules (fatty acids and cytokines) that impact blood and vessel cells, inducing low-grade systemic inflammation that can transition to tissues, including the brain. Inflammation in the brain—neuroinflammation—is one of key elements of the pathobiology of neurodegenerative disorders; it is characterized by the activation of microglia, the resident immune cells in the brain, and by the structural and functional changes of other cells forming the brain parenchyma, including neurons. Such cellular changes have been shown in animal models with direct methods, such as confocal microscopy. In humans, cellular changes are less tangible, as only indirect methods such as magnetic resonance (MR) imaging are usually used. In these studies, obesity and low-grade systemic inflammation have been associated with lower volumes of the cerebral gray matter, cortex, and hippocampus, as well as altered tissue MR properties (suggesting microstructural variations in cellular and molecular composition). How these structural variations in the human brain observed using MR imaging relate to the cellular variations in the animal brain seen with microscopy is not well understood. This review describes the current understanding of neuroinflammation in the context of obesity-induced systemic inflammation, and it highlights need for the bridge between animal microscopy and human MR imaging studies.

## 1. Introduction

Obesity is a major risk factor of Alzheimer’s disease and related dementias (ADRD) [1,2]. It is common, affecting >40% of the adult US population [3], and it increases the risk by 1.6-fold [1,2]; consequently, >13% of the ADRD incidence in the population is attributable to obesity [1,2]. Considering that the prevalence of obesity is still rising [3], this number is expected to increase in the near future. Dementia has a long prodromal phase, developing progressively over decades before the clinical diagnosis [4,5,6]. An early identification of individuals at risk and the implementation of effective strategies for prevention and early treatment are essential [2], as delaying symptom onset by just 1 year could lower the prevalence of ADRD by >9 million cases in the USA in the next 40 years [2].

The principal feature of dementia is the loss of neurons and brain atrophy [7,8,9]. The mechanistic links between obesity and the neurodegenerative processes of dementias are not fully understood, but several lines of evidence suggest that obesity-related systemic inflammation and subsequent neuroinflammation may be involved [10,11,12]. Adipose tissue adipocytes and macrophages release multiple proinflammatory molecules [13,14,15,16] that enter the systemic circulation, where they impact blood cells and endothelial cells lining the vessels, and promote the transition of this low-grade inflammation from the circulation to tissues, including the brain. 

Inflammation in the brain—*neuroinflammation*—is one of the key elements of the pathobiology of neurodegenerative disorders [17,18,19]. On a cellular level, neuroinflammation is characterized by the activation of microglia, resident immune cells in the brain. Microglia alter their structure and function in response to the pro-inflammatory signals from the circulation; other cells forming the brain parenchyma, including astrocytes, oligodendrocytes, and neurons, also alter their structure and function in response to pro-inflammatory signals [20,21,22,23,24,25,26,27,28,29,30,31]. Collectively, neuroinflammation creates a tissue environment that becomes less neurotrophic and more pro-apoptotic; in the long-term, this may contribute to neurodegeneration. Most of these observations have been made in animal studies using *microscopy imaging*, which has the capacity to examine detailed cellular variations (at ‘micrometer’ to ‘nanometer’ resolutions) in (mostly) pre-selected regions of the brain. Such studies are much less feasible in humans, as microscopy imaging is (largely) limited to post mortem samples, which are sparse. In vivo studies of the human brain are mainly conducted with *magnetic resonance (MR) imaging*, which has the capacity to assess volumetric variations of neuroanatomical structures (at ‘millimeter’ resolution), and variations in tissue MR properties across the whole brain [32]. These human MR imaging studies have shown that adiposity is associated with variations in volumes, as well as in tissue MR-based properties. How these structural variations in the human brain observed with MR imaging relate to the cellular variations in the animal brain seen with microscopy is not well understood. In an attempt to start bridging this gap, here, we will *first* describe the key findings of human studies of the brain in obesity, as assessed with in vivo MR imaging, and *second,* we will describe the main observations in animal studies of the brain in obesity, as examined with ex vivo microscopy imaging. We will begin each part with some basic methodological considerations. We will focus this minireview on the cerebrum, which is the largest part of the brain and which exhibits changes during the preclinical and clinical course of AD (leading ultimately to brain atrophy) [7,8,9,33]. Note that the hypothalamus is outside the scope of this minireview, as it has been reviewed in the context of obesity previously [34].

## 2. MR Imaging of the Brain in Obesity

### 2.1. Methodological Considerations: Basic Principles and Modalities Employed

#### 2.1.1. Basic Principles

MR imaging exploits the magnetization properties of hydrogen nuclei within the imaged tissue, such as the brain. Most often, MR imaging is based on radiofrequency signals generated by hydrogens, which are present in great abundance in biological tissues and have a high gyromagnetic ratio that makes the signal strong. The nucleus of hydrogen consists of one proton. Hydrogen protons are charged particles normally spinning with random orientation. In an MR scanner, a strong (static) magnetic field is present (e.g., 3.0 T), and hydrogen protons become aligned with this magnetic field (i.e., they become magnetized in the longitudinal direction). The direction of magnetization can be changed (or flipped) by an external radio frequency (RF) energy (RF pulse). The angle at which the magnetization is changed (e.g., 90°) or flipped (e.g., 180°) is determined by the duration and strength of the RF pulse. For example, a RF pulse inducing transverse magnetization is when the direction of tissue magnetization is at a 90° angle with respect to the direction of the magnetic field. When the RF pulse is terminated, hydrogen protons return to the original alignment with the magnetic field of the scanner through various relaxation processes, during which they emit RF energy. After a certain period following the initial RF, the emitted signals are measured. Different types of images can be created by different sequences of RF pulses. These sequences are characterized by repetition time (TR), which is the time between successive RF pulses applied, and time to echo (TE), which is the time between the delivery of the RF pulse and the receipt of the echo signal. The relaxation processes through which excited hydrogen protons realign with the magnetic field of the scanner are characterized by T1- and T2-relaxation times. The T1-relaxation time is the time in which the longitudinal magnetization re-grows (after the RF pulse). The T2-relaxation time is the time in which the transverse magnetization dissipates (after the RF pulse). The relaxation processes differ for hydrogen protons in water molecules in extracellular and intracellular spaces (characterized by higher molecular motion) vs. hydrogen protons in macromolecules in semi-solid cell structures (characterized by lower molecular motion). This phenomenon is key in creating *tissue contrasts on MR images*, visualizing brain structures differing by molecular composition (high content of water (e.g., cerebrospinal fluid) vs. macromolecules (e.g., fatty myelin and lipid composition of cell membranes)).

#### 2.1.2. Modalities Employed

*(i) T1-weighted (T1W) imaging* uses sequences with short TE and TR, and the signal is mainly determined by the T1-relaxation properties. This imaging modality is commonly used to assess volumes, such as those of the whole brain and its different subregions, including regions the cerebral cortex or the hippocampus (Figure 1). *(ii) T2-weighted (T2W) imaging* uses sequences with longer TE and TR times, and the signal is mainly determined by the T2-relaxation properties; this modality is often used to visualize small lesions in white matter, including white matter hyperintensities, and quantify their volume. *(iii) T2W Fluid Attenuated Inversion Recovery (FLAIR) imaging* uses sequences that are similar to those of T2W imaging, except that the TE and TR times are longer. This modality is also often utilized to visualize and quantify white matter hyperintensities. *(iv) Diffusion tensor imaging (DTI)* is a T2-relaxation time-based imaging technique that assesses the diffusion of water molecules in tissue. It assesses the quantity and directionality of water diffusion within so-called diffusion tensor (conceptualized as a three-dimensional ellipsoid). The diffusion tensor is described using eigenvectors and eigenvalues, with the largest eigenvector being oriented in parallel with neuronal axons. DTI can provide insights into the architecture and integrity of myelinated axons, and the fiber tracts that these axons form. In myelinated axons, water molecules move along the axonal tracts, whereas in axons with disrupted myelin, water molecules can move also radially [35]. The main measures of DTI (characteristics of diffusion tensors) are fractional anisotropy (FA), radial diffusivity (RD), and mean diffusivity (MD). In general, decreased FA, increased RD, and increased MD are considered as indicators of myelin disruption and axonal damage. *(v) Magnetization transfer ratio (MTR) imaging* is also a T2-relaxation time-based modality; it generates contrast on MR images using the process of magnetization transfer between hydrogen protons bound to water and hydrogen protons bound to macromolecules [36,37,38]. Magnetization transfer is facilitated by the difference in molecular motion between the two pools of hydrogen protons [39]. In white matter, MTR is commonly assumed to index myelin content and structural properties [40], but, especially in gray matter, it may also index variations in lipid content within cell membranes [41,42] or cytosol (e.g., lipid droplets). *vi) Normalized T1W signal intensity (nT1W-SI)* is a measure derived using the T1W MR imaging technique described above; it is determined as the signal intensity of a brain region of interest normalized by the signal intensity of the whole brain [43]. This metric may also index variations in myelin and lipid content within the cell membranes [41,42] and cytosol (e.g., lipid droplets).

### 2.2. Findings

Before we describe the MR imaging characteristics of the brain in obesity, we will do so for a typical brain.

#### 2.2.1. MR Structure of a Typical Brain

On a neuroanatomical MR image (Figure 1), the cerebrum consists of grey and white matter, with the grey mater including the cerebral cortex, which is a 2 to 4 mm thick outer layer of the cerebrum, and individual subcortical structures, such as the hippocampus [44], and the white matter, which is located underneath the cortex, including structures such as the corpus callosum [44].

#### 2.2.2. MR Structure of the Brain in Obesity

A growing body of research over the last decade has shown that higher body adiposity (and/or systemic inflammation) is associated—mostly cross-sectionally—with variations in brain structure. Much of this research has used body mass index (BMI) as an index of adiposity (body weight in kg/height in m^2^), which is an indirect measure of adiposity determined not only by fat mass, but also by muscle and bone mass. Only a few studies have examined body fat more directly (with MRI or computerized tomography), and have quantified the amount of visceral fat (i.e., intra-abdominal fat) [43,45,46,47,48,49]. To assess visceral vs. non-visceral fat (mostly subcutaneous fat, i.e., the fat under the skin) may be important, as visceral fat (vs. subcutaneous fat) produces more proinflammatory molecules [21,50,51,52,53], and individuals with a normal BMI but high visceral fat are at higher risk of developing systemic inflammation [50]. 

The adiposity–brain structure research has been reviewed in detail elsewhere [54]. Here, we highlight only the main findings. Thus, using T1W imaging, most studies have shown that a higher adiposity is associated with a lower brain volume, lower cortical grey volume, and/or lower cortical thickness [45,54,55,56,57,58], and that the associations with cortical thickness are particularly strong in the frontal, temporal, and parietal cortices [54,56]. A higher adiposity is also associated with lower volumes of the hippocampus [57]. Using T2W and T2-FLAIR imaging, a higher adiposity is associated with higher volumes of white matter hyperintensities [59], which are a marker of small vessel disease [59]. Regarding tissue MR properties, several studies have also shown that a higher adiposity is associated with altered white matter properties, as indicated by lower FA and higher MD on DTI [56,60,61], and higher nT1WSI and MTR [43,47,62]. 

Some of the above studies suggest that the adiposity–brain structure relationships are stronger for visceral fat (accumulating intra-abdominally) than for fat stored elsewhere in the body (mainly subcutaneously) [45,62]. Some studies also suggest that at least a part of the observed adiposity–brain structure relationships may be mediated by systemic inflammation [45,47,62]. Further, it is becoming evident that the adiposity–brain structure relationships do only not exist during adulthood, but also during childhood and adolescence [63], and that they vary across the lifespan [64] and length of exposure (cumulating effects of sustained obesity) [65]. Further, excess adiposity induces not only systemic inflammation, but also metabolic abnormalities, such as insulin resistance and dyslipidemia [66,67], which may potentiate the effects of and/or contribute to the development of neuroinflammation. Thus, e.g., the proinflammatory cytokine TNFα, the production of which is increased by ‘obese’ adipose tissue, plays a key role in the development of insulin resistance [68], which exists not only in the periphery, but also in the brain [69], where it has been associated with lower cortical thickness in the regions that overlap with those demonstrating a lower thickness in association with higher adiposity [70,71,72,73,74,75,76]. Regarding dyslipidemia, a higher adiposity increases the blood concentrations of fatty acids that are enriched in saturated and monounsaturated fatty acids, which can directly stimulate neuroinflammation [67], and that are depleted in polyunsaturated fatty acids, which are essential building blocks of neuronal cell membranes and may inhibit neuroinflammation [77,78]. Consistently, higher blood levels of polyunsaturated fatty acids (incorporated in circulating triacylglycerols) are associated with higher cortical thickness [79]. Finally, many of the observed adiposity–brain structure relationships have been associated with impaired cognitive functioning [80]. 

Of note, the adiposity–brain structure relationships may exist not only in the direction from obesity to brain structure but also in the direction from brain structure to obesity (via, e.g., the brain regulation of eating behavior [81,82], reviewed recently elsewhere [54]). 

In contrast to human research, only a few studies have investigated adiposity–brain structure relationships using brain MR imaging in animal models (Figure 1). Thus, in one mouse study of diet-induced obesity (in response to chronic high-fat diet feeding), a higher adiposity was associated with lower volumes of the frontal, temporal, and parietal cortices [83]. In another mouse study of diet-induced obesity, a higher adiposity was associated with higher MD on DTI of the cerebrum [84].

## 3. Microscopy Imaging of the Brain in Obesity

### 3.1. Methodological Considerations: Basic Principles and Modalities Employed

#### 3.1.1. Basic Principles

Several types of microscopy can be employed to image brain cells and their organelles. Most commonly, these include *optical (light) microscopy* [85], which involves magnifying the image of the object (a tissue sample) by passing light (photons) through it or reflecting light off it, and then examining this light through a single or multiple lens. Less commonly used types of microscopy include *electron microscopy*, which utilizes an electron beam instead of light, and electromagnets instead of traditional lenses; as the electron beam has a far smaller wavelength than light, the image resolution can increase from micrometers to nanometers [86]. 

Regarding optical microscopy, a variety of methods exist, with fluorescence microscopy being one of the most often used, and it is undergoing, at present, major advancements that are aimed to increase the speed of image acquisition, image resolution, and the volume of imaged tissue. Fluorescence microscopy exploits the capacity of some molecules to fluoresce, that is, to absorb and emit light. For the purpose of microscopy, such fluorescent molecules (probes) can be attached to (or integrated into) different cell structures and imaged. In a confocal microscope, which is the work horse of current fluorescence microcopy, the illumination and detection optics are focused on the same diffraction-limited spot in the sample, which is the only spot imaged by the detector during a confocal scan (while out-of-focus light is rejected), allowing for high-resolution image acquisition in thick tissues [87,88]. To generate a complete image, the spot is moved over the sample, and data are collected point by point (*laser-point scanning*) [87]. Acquiring high-resolution stacks of such images (*optical sectioning*) enables 3D reconstructions of tissue samples [87]. A higher speed of image acquisition is achieved by more recently developed light-sheet fluorescence microscopy [89], which illuminates a thin laminar volume (with a laminar laser beam, termed a light sheet) rather than a single point. A further reduction of out-of-focus light and greater tissue penetration is achieved with two-photon microscopy, which uses two low-energy excitation photons rather than one high-energy excitation photon [90]. 

These microscope developments have been complemented by substantive innovations in sample preparation, such as *optical tissue clearing* (OTC) [91,92,93]. OTC methods remove lipids (delipidation), pigments (decolorization), and calcium phosphate (decalcification), and they aim to match the refractive indices of the tissue sample and imaging media, reaching an almost complete level of transparency [91]. This reduces light scattering (and thus optical background) and, as such, allows for the acquisition of high-resolution images of large sections, or even the whole rodent brain [94,95,96]. Currently, hydrogel-based tissue-clearing methods [94,97,98,99] are becoming the most commonly used methods. They include ‘cleared lipid-extracted acryl-hybridized rigid immunostaining/in situ hybridization-compatible tissue hydrogel’ (CLARITY) [100] and ‘stabilization to harsh conditions via intramolecular epoxide linkages to prevent degradation’ (SHIELD) [101]. CLARITY and SHIELD allow for a uniform delipidation of the tissue, and due to the presence of the hydrogel, also the stabilization and preservation of a large assortment of cell biomolecules through covalent linkages [98]. These preserved tissues can then withstand multiple rounds of antibody staining, stripping, and reprobing, vastly increasing the amount of cellular level information that can be gained from a single sample. 

#### 3.1.2. Modalities Employed

Microscopy methods applied to the brain in obesity include: *(i) electron microscopy* for assessing disruption in myelin membranes [29], *(ii) bright-field light microscopy* for evaluating neuronal dendritic spine density [20], *(iii) confocal fluorescence microscopy* for assessing cell-body size and ramifications of microglia [20,28], and *(iv) light-sheet microscopy with tissue clearing* for examining the blood–brain barrier across the whole brain for uptake of peripherally administered leptin [102].

### 3.2. Findings

Before we describe observations from microscopy imaging of the brain in obesity, we will describe the cellular composition of the cerebral parenchyma for a typical brain. 

#### 3.2.1. Brain Cells in a Typical Brain

The human brain contains approximately 85 billion neurons and about the same number of glial cells [103]. Of the glial cells, ~10% or less are microglia, 19–40% are astrocytes, and 45–75% are oligodendrocytes [104] (Figure 2). *Neurons* are the principal cells of the brain; their degeneration and loss are the primary features of AD and related dementias [105]. A typical neuron consists of a cell body, multiple branching dendrites (dendritic arbor), and a single axon; dendrites have small projections, spines. Axons form a network that totals 176,000 km in length in the adult brain [106]. The speed of conduction along the axon depends on two of its structural properties: axon diameter and myelin sheath. The former is determined by an axonal cytoskeleton constituted by neurofilaments, and the latter insulate axons along their length, except for the nodes of Ranvier, for fast (saltatory) conduction of action potentials [107]. The great majority of axons are short (<3 mm, intra-cortical axons); longer (3–30 mm) and very long (>30 m) axons exit (efferent) and enter (afferent) the cortex, forming the bulk of white matter [108]. Grey matter contains neuronal cell bodies, dendrites, and short axons (and some parts of the longer ones). *Microglia* are the resident immune cells of the brain. They are cells with multiple ramified processes that continuously survey the surrounding parenchyma [109]. When microglia detect cellular debris, apoptotic cells, foreign material, or pro-inflammatory cytokines, they undergo graded ‘activation’, during which their cell bodies enlarge and their ramified processes shorten [110]. ‘Resting’ microglia are responsible for scavenging, whereas ‘activated’ microglia are responsible for cytotoxicity, antigen presentation, and tissue repair. Both resting and activated microglia have the capacity for phagocytosis and extracellular signaling [111]. *Astrocytes* are glial cells with multiple supporting actions. In gray matter, astrocytes have short and thick primary branches that divide into thinner secondary branchlets and tertiary leaflets. The leaflets have characteristic endfeet that encase blood vessels and ensheath thousands of synapses; the processes of a single astrocyte can contact up to 2 million synapses in the human brain [112]. In white matter, astrocytes have long, thin, and mostly unbranched processes [113] that are aligned with axons, mostly at the nodes of Ranvier [114]. Astrocytes support neurons by releasing neurotrophic factors [115]. They also store and supply energy to neurons—they take up glucose to synthesize and store glycogen, which is used later as an energy source when needed [116]. Astrocytes play a major role in the coupling of neuronal activity to blood flow [117,118,119], and they participate in the formation and elimination of synapses [120]. Astrocytes maintain the fluid and pH homeostasis of the synaptic interstitial fluid via the expression of water channels and ion transporters [121]. *Oligodendrocytes* are glial cells that form myelin sheaths, provide trophic support to neurons, and modulate neuronal excitability [122]. Myelin sheaths are essentially cell membranes of multiple oligodendrocyte processes that are tightly packed (40 or more lipid bilayers) around neuronal axons [123,124]. Myelin sheaths are lipid-rich, containing 70–85% lipids and only 15–30% proteins (which is unlike other cell membranes, which contain approximately equal amounts of lipids and proteins) [125]. The brain parenchyma is separated from the blood circulation by *the blood–brain barrier* (Figure 2), which consists of endothelial cells, pericytes, and astrocytic endfeet. The blood–brain barrier regulates the transport of molecules (and immune cells) to the brain parenchyma [126].

#### 3.2.2. Brain Cells in Obesity

Based mainly on preclinical studies of diet-induced obesity (in mice and rats) in selected brain regions (e.g., parts of the cerebral cortex or hippocampus), obese (vs. lean) animals show neurons with reduced spine density and disrupted myelin, and a higher presence of activated microglia and reactive astrocytes (both with enlarged soma size and reduced ramifications) [20,22,25,26,27,28,29,30,31] (Figure 2). These cellular changes may develop due to the release of—by an expanding adipose tissue—multiple pro-inflammatory molecules (saturated and mono-unsaturated FAs and cytokines, such as IL-6 and TNFα) into the circulation, where they generate a hyperlipidemic, pro-inflammatory milieu that disrupts the functional and structural integrity of the blood–brain barrier; consequently, more lipids, proinflammatory molecules, and immune cells enter the brain parenchyma, and promote the development of neuroinflammation [26,27,127,128]—microglia and astrocytes become more proinflammatory (and cytotoxic) and less homeostatic (and neurotrophic). Both microglia and astrocytes increase the production of proinflammatory cytokines and reactive oxygen species, which impair mitochondrial function, damage DNA, proteins, and cell membranes, and promote apoptosis in surrounding cells, including neurons and oligodendrocytes; at the same time, they reduce the phagocytosis of cell and myelin debris and the production of neurotrophic factors that are required for the differentiation of oligodendrocytes and new myelin production, and for neuron survival [129,130,131,132,133,134]. Finally, in obesity and under inflammatory conditions, microglia, astrocytes, and neurons accumulate lipid droplets (LDs) in their cytoplasm [135,136]. Among others [137,138,139,140], LDs sequester excess lipids from the cytoplasm to prevent lipotoxicity, as excess lipids can disrupt the integrity of cell membranes [141]. Reactive oxygen species upregulate lipid synthesis in neurons that transfer excess fatty acids to astrocytes, where they are absorbed by LDs [142,143]. Lipid-laden astrocytes produce proinflammatory factors, such as IL-6 and TNFα [135,136]. Under inflammatory conditions, microglia accumulate LDs [142,144], which are accompanied by compromised phagocytic function. Thus, a number of cellular brain changes have been observed in obesity, but how these changes relate to the observed volumetric and tissue MR properties remains to be elucidated. 

## 4. Conclusions and Future Directions

Human brain MR imaging studies show that a higher body adiposity is associated with lower volumes of the cerebral grey matter (including the cortex and hippocampus) and the alteration of tissue MR properties of the cerebral white matter. Brain MR imaging studies in animals are sparse, but the few that have been conducted provide similar results. Brain microscopy in animal studies shows that diet-induced obesity results in morphological changes of multiple cell types, including microglia, astrocytes, oligodendrocytes, and neurons, but how these cellular changes seen with microscopy relate to the changes in volumes and tissue MR properties observed with MR imaging is not clear at present. Clearly, there is a need for a bridge between animal microscopy and human MR imaging studies. Fast advancing microscopy (with tissue clearing and multiplex immunohistochemistry) that has the capacity to image the entire rodent brain, combined with MR imaging, may provide such a bridge. Towards that end, some novel approaches have been developed to infer cell biology behind the MR imaging data by combining such data with in silico data available in mRNA and protein brain atlases [70,79,145].

## Figures and Tables

**Figure 1 ijms-23-08790-f001:**
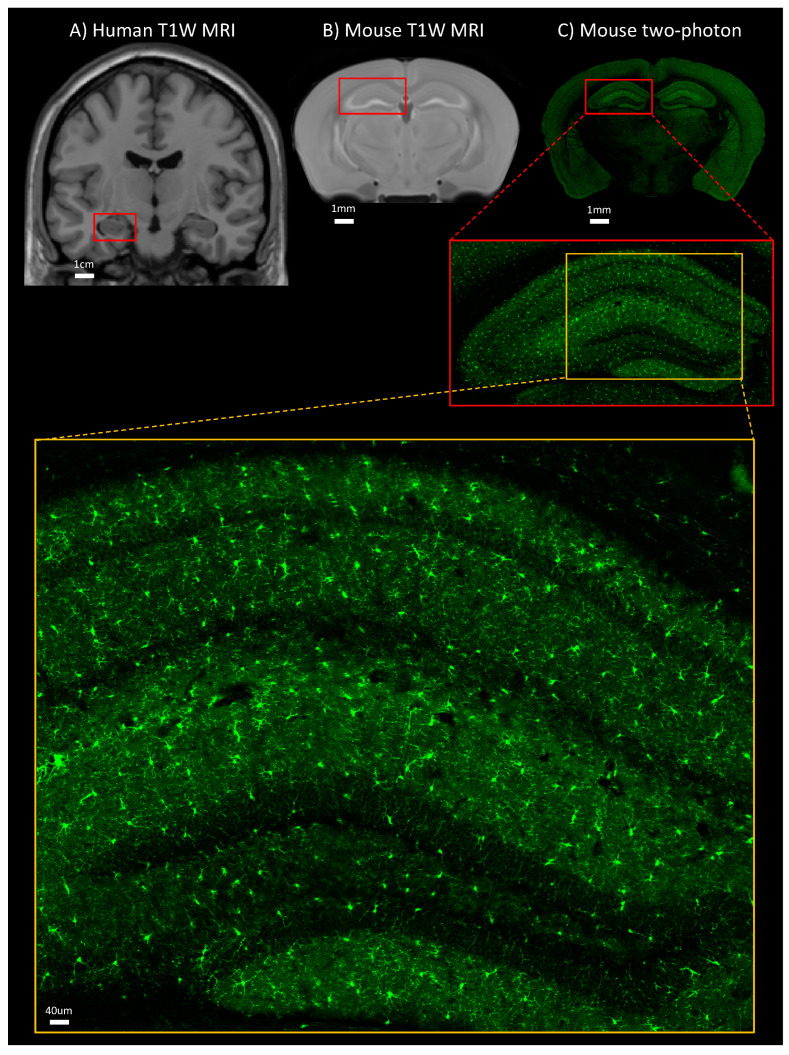
Coronal sections of the brain: (**A**) T1W MR image in humans, (**B**) T1W MRI in mice, and (**C**) two-photon microscopy in microglia-reporter mouse labeled with GFP. Red and yellow rectangles indicate the hippocampus and its subsections. T1W: T1-weighted, MRI: magnetic resonance imaging, GFP: green fluorescent protein. Unpublished data are shown.

**Figure 2 ijms-23-08790-f002:**
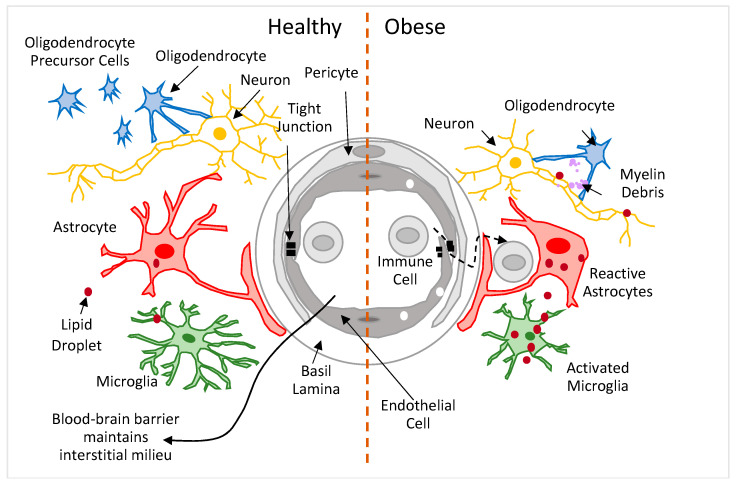
Schematic of cellular composition of brain tissue in healthy and obese brains.

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
