# Peer review of "Obesity-Related Neuroinflammation: Magnetic Resonance and Microscopy Imaging of the Brain"

_ijms, 2022, doi:10.3390/ijms23158790_

Round 1

Reviewer 1 Report

The manuscript may benefit to include a more images of comparisons of human vs mouse brain. Especially, during the disease states to indicate the differences between both.

Author Response

Reviewer 1
The manuscript may benefit to include a more images of comparisons of human vs mouse brain. Especially, during the disease states to indicate the differences between both.

Response: To the best of our knowledge, we are not aware of any additional suitable images of human vs. mouse brain that would indicate species differences relevant to obesity. 

Reviewer 2 Report

This review focuses on the techniques used to observe the structural and cellular variations in the brain cells under obesity, and the link with neuroinflammation which is an important pathological event in Alzheimer’s disease and related dementias. The findings from this study may shed light on future research and diagnosis of obesity-induced dementia. I recommend considering the following points before acceptance. 

Major: Obesity is shown to associate with neuroinflammation, particularly in the hypothalamus, and this is partially responsible for these negative cognitive outcomes. What are the current methodological considerations and modalities employed to observe the change of this particular area in the brain under obesity?

Minor: The font in the manuscript is not unified, sometimes italics sometimes underlined. Not sure if that is appropriate

Table 1 forms a paragraph, but not a table? The font of the text is smaller.

Is Figure 1 cited from a reference? If so, please include the citation. Please improve the resolution of the graph, as the text in the graph is blurry and hard to read; RFP: red fluorescent protein seems not visible

Both Figure 2 and its legend are blurry; the legend is repeated

In section 1.2.2, how does obesity induce the release of multiple pro-inflammatory molecules associated with adipose tissue? 

Author Response

Reviewer 2:
This review focuses on the techniques used to observe the structural and cellular variations in the brain cells under obesity, and the link with neuroinflammation which is an important pathological event in Alzheimer’s disease and related dementias. The findings from this study may shed light on future research and diagnosis of obesity-induced dementia. I recommend considering the following points before acceptance. 

1. Major: Obesity is shown to associate with neuroinflammation, particularly in the hypothalamus, and this is partially responsible for these negative cognitive outcomes. What are the current methodological considerations and modalities employed to observe the change of this particular area in the brain under obesity?

The research on the hypothalamus in obesity is outside the scope of this minireview, as it has been reviewed in detail previously (PMID: 28045396). We make the readership aware of this point as follows on page 3: “We will focus this minireview on the cerebrum, which is the largest part of the brain and which exhibits changes during the preclinical and clinical course of AD (leading ultimately to brain atrophy)7-9, 32. Note that the hypothalamus is outside the scope of this minireview, as it has been reviewed in the context of obesity previously33.”

2. Minor: The font in the manuscript is not unified, sometimes italics sometimes underlined. Not sure if that is appropriate

This has been corrected throughout the manuscript.

3. Table 1 forms a paragraph, but not a table? The font of the text is smaller.

This was a typo made by the IJMS editors when re-formatting our manuscript to fit the format of the journal. There is no table in this manuscript. “Table 1” was supposed to read “T1-weighted”. This has now been corrected. The font of the text has been enlarged.

4. Is Figure 1 cited from a reference? If so, please include the citation. Please improve the resolution of the graph, as the text in the graph is blurry and hard to read; RFP: red fluorescent protein seems not visible.

-No, Figure 1 is not cited from a reference; it shows our unpublished data, which is now noted in the figure legend. 
-We have now included a higher-resolution image of the figure. 
-The text in the figure was a figure legend, which we have deleted from the figure, as it is now placed below the figure.
-RFP: red fluorescent protein is not shown in the images; it is no longer mentioned the figure legend.

5. Both Figure 2 and its legend are blurry; the legend is repeated

-As above, we have included a higher-resolution image of the figure now.
-The duplicated figure legend has been deleted.

6. In section 1.2.2, how does obesity induce the release of multiple pro-inflammatory molecules associated with adipose tissue? 

To address this question, we expanded the second paragraph of Introduction on page 3 as follows:
“Adipose tissue adipocytes and macrophages release multiple proinflammatory molecules13-15 that get into the systemic circulation where they impact blood cells and endothelial cells lining the vessels and promote transition of this low-grade inflammation from the circulation to tissues, including the brain.”. 

In this paragraph, we also provided an additional reference on adipose tissue as an endocrine secretory organ (DOI:10.1079/PNS200194).

Further, in section 2.2.2. on page 8, we write that: “These cellular changes may develop due to the release of – by an expanding adipose tissue – multiple pro-inflammatory molecules (saturated and mono-unsaturated FAs and cytokines, such as IL-6 and TNFα) into the circulation where they generate a hyperlipidemic, pro-inflammatory milieu that disrupts the functional and structural integrity of the blood-brain barrier;”.
